

# Vocal repertoire of *Microhyla nilphamariensis* from Delhi and comparison with closely related *M. ornata* populations from the western coast of India and Sri Lanka

Megha Srigyan[1,2,3,*], Abdus Samad[1,4], Abhishek Singh[1,4], Jyotsna Karan[1,4], Abhishek Chandra[1], Pooja Gokhale Sinha[5], Vineeth Kumar[6], Sandeep Das[7,8], Ashish Thomas[9] and Robin Suyesh[1,*]

[1] Department of Environmental Sciences, Sri Venkateswara College, University of Delhi, New Delhi, Delhi, India

[2] Department of Biochemistry, Sri Venkateswara College, University of Delhi, New Delhi, Delhi, India

[3] Department of Ecology and Evolutionary Biology, University of California, Santa Cruz, Santa Cruz, CA, United States of America

[4] Biological Sciences, Sri Venkateswara College, University of Delhi, New Delhi, Delhi, India

[5] Department of Botany, Sri Venkateswara College, University of Delhi, New Delhi, Delhi, India

[6] Department of Biology, Center for Advanced Learning, Mangalore, Karnataka, India

[7] Forest Ecology and Biodiversity Conservation Division, Kerala Forest Research Institute, Peechi, Kerala, India

[8] Department of Zoology, St Joseph's College (Autonomous), Irinjalakuda, Thrissur, Kerala, India

[9] Department of Environmental Studies, SGND Khalsa College, University of Delhi, Delhi, India

[*] These authors contributed equally to this work.

Corresponding author
Robin Suyesh, robins@svc.ac.in

## ABSTRACT

Advertisement calls in frogs have evolved to be species-specific signals of recognition and are therefore considered an essential component of integrative taxonomic approaches to identify species and delineate their distribution range. The species rich genus *Microhyla* is a particularly challenging group for species identification, discovery and conservation management due to the small size, conserved morphology and wide distribution of its members, necessitating the need for a thorough description of their vocalization. In this study, we provide quantitative description of the vocal behaviour of *Microhyla nilphamariensis*, a widely distributed south Asian species, from Delhi, India, based on call recordings of 18 individuals and assessment of 21 call properties. Based on the properties measured across 360 calls, we find that a typical advertisement call of *M. nilphamariensis* lasts for 393.5 $\pm$ 57.5 ms, has 17 pulses on average and produce pulses at rate of 39 pulses/s. The overall call dominant frequency was found to be 2.8 KHz and the call spectrum consisted of two dominant frequency peaks centered at 1.6 KHz and 3.6 KHz, ranging between 1.5–4.1 KHz. Apart from its typical advertisement call, our study also reveals the presence of three 'rare' call types, previously unreported in this species. We describe variability in call properties and discuss their relation to body size and temperature. We found that overall dominant frequency 1 (spectral property) was found to be correlated with body size, while first pulse period (temporal property) was found to be correlated with temperature. Further, we compare the vocal repertoire of *M. nilphamariensis* with that of the congener *Microhyla ornata* from the western coast

of India and Sri Lanka and also compare the call properties of these two populations of *M. ornata* to investigate intra-specific call variation. We find statistically significant differentiation in their acoustic repertoire in both cases. Based on 18 call properties (out of 20), individuals of each locality clearly segregate on PCA factor plane forming separate groups. Discriminant function analysis (DFA) using PCA factors shows 100% classification success with individuals of each locality getting classified to a discrete group. This confirms significant acoustic differentiation between these species as well as between geographically distant conspecifics. The data generated in this study will be useful for comparative bioacoustic analysis of *Microhyla* species and can be utilized to monitor populations and devise conservation management plan for threatened species in this group.

## INTRODUCTION

Quantitative descriptions of advertisement calls serve a crucial role in understanding the diverse vocal repertoire exhibited by anurans. In most species, these calls are produced by males to attract gravid conspecific females and indicate calling territories to rival males (*Wells, 2007*). Such signals are subject to sexual selection and function in aspects like mate choice, body size assessment, fitness and competition within a species (*Davies & Halliday, 1978*; *Ryan & Keddy-Hector, 1992*; *Welch, 1998*). Detailed acoustical and statistical descriptions of such calls are thus necessary to understand how vocal signals shape social and reproductive behaviour in anurans (*Kelley, 2004*; *Gerhardt, 2006*; *Wells, 2007*; *Bee et al., 2010*). Quantitative bioacoustic descriptions have become paramount for taxonomic revisions, which are on the rise as identification methods become more accurate and robust (*Dayrat, 2005*; *Vieites et al., 2009*; *Glaw et al., 2010*; *Köhler et al., 2017*). Since vocalizations are species-specific, they provide an unambiguous method to identify and discern cryptic species from one another (*Narins et al., 1998*; *Padial & De La Riva, 2009*; *Klymus et al., 2010*; *Angulo & Icochea, 2010*; *Jansen et al., 2011*). Additionally, the data generated may be used in conservation efforts to develop non-invasive population monitoring and biodiversity management strategies against alarming amphibian declines (*Bridges & Dorcas, 2000*; *Stuart et al., 2004*; *Hsu, Kam & Fellers, 2005*; *Laiolo, 2010*; *Blumstein et al., 2011*).

The genus *Microhyla* Tschudi, 1838 consists of an assemblage of small sized, narrow mouthed, ground dwelling frogs having conserved morphology and wide distribution (*Garg & Biju, 2019*). The group currently comprises 50 species distributed throughout South, South-east and East Asia (*Frost, 2021*; *Gorin et al., 2020*). Due to its wide distribution, conserved morphology, limited sampling and varying levels of intra- and inter-specific genetic divergence, accurate species identification has remained challenging in this genus (*Seshadri et al., 2016*; *Garg & Biju, 2019*). Despite the remarkable surge in studies reporting novel Microhylids in recent years (*Garg & Biju, 2019*), it is suggested that species diversity

in this group still remains highly underestimated (*Gorin et al., 2020*). Clearly, an integrative taxonomic approach that involves descriptions of vocal repertoire is needed to resolve many of these aspects (*Vences et al., 2010*; *Wijayathilaka et al., 2016*; *Garg & Biju, 2019*; *Garg & Biju, 2019*; *Garg et al., 2021*).

Our study species, *M. nilphamariensis,* was first reported from Bangladesh (*Howlader et al., 2015*) and subsequently from Nepal (*Khatiwada et al., 2017*), although it was often reported previously in literature as the widespread *M. ornata* (*Hasan et al., 2012*; *Hazra, 2016*; *Vineeth et al., 2018*) which was thought to be distributed widely in the Indian subcontinent. Recent molecular and phylogenetic work assigned *M. nilphamariensis* to the '*M. ornata* species complex' consisting of four closely allied members (*Garg et al., 2018*; *Garg & Biju, 2019*; *Gorin et al., 2020*). According to the now revised distribution, *M. nilphamariensis* is widespread across India, ranging from north/north-east, through central India and as far south as northern regions of Karnataka in addition to Bangladesh and Nepal, making it the most widespread member of the *Microhyla* genus in South Asia (*Garg et al., 2018*). However, descriptions of vocal repertoire of this species, a vital component of integrative taxonomic approach, still remain highly limited (*Hasan et al., 2015*; *Vineeth et al., 2018*; *Garg & Biju, 2019*). This necessitates extensive sampling of call recordings from different localities to understand its vocal repertoire across its distribution range. In addition to descriptions of vocal behaviour of a particular species from an area, comparisons with closely related congeners distributed across different regions are important in identifying underlying patterns of acoustic differences between them (*Littlejohn & Oldham, 1968*; *Klymus et al., 2010*). Since *M. nilphamariensis* and *M. ornata* have been shown to be phylogenetically closely related species having overlapping distribution range in some areas (*Garg et al., 2018*; *Garg et al., 2019*) a comparative analysis of vocal behaviour between them is an excellent opportunity to investigate the divergence of acoustic traits and the processes driving them between such interrelated species.

In this study, we provide a comprehensive quantitative description of the advertisement calls of *Microhyla nilphamariensis* from Delhi, India and discuss the patterns and sources of call variation, both within and among individuals. We also compare the vocal repertoire of *Microhyla nilphamariensis* with that of its congener, *Microhyla ornata,* from the western coast of India and Sri Lanka. We further analyse the divergence of vocal repertoire between two populations of *M. ornata*, one from the western coast of India and the other from Sri Lanka, to investigate intra-specific variations in call characteristics. Overall this study will be instrumental to reveal interspecific and intraspecific patterns of acoustic differences, which will encourage further investigations into their taxonomy, genetic diversity and acoustic signaling pattern in this group of frogs.

## MATERIALS AND METHODS

### Calling site

Calling males of *Microhyla nilphamariensis* were recorded between 28 August 2015 to 15 October 2015 from a population found in the Central Ridge Forest, New Delhi, India (28.5886°N, 77.1607°E, alt 248 m asl; Fig. 1). The Ridge is an important element of the

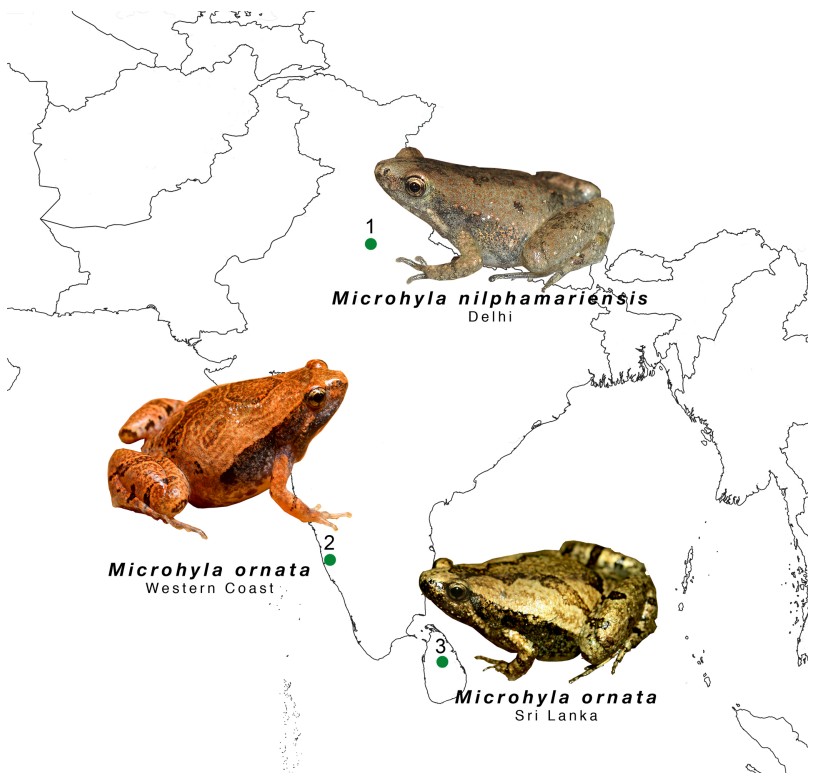

**Figure 1** **The study sites of the of two microhyla species.** *Microhyla nilphanariensis* and *Microhyla ornata* from the Indian sub-continent.

region's physiography and covers a length of nearly 35 km in various fragments across the city, hosting a variety of flora and fauna, including anurans. The landscape is mostly a dry deciduous shrub forest and is adjacent to traffic-heavy roads. The study population was located near a seasonal pond lined with weeds and tall grass. Individuals were mostly found to call in grass and leaf litter, up to distances ranging from 0.5 to 20 m from the pond. Calling activity was observed to be synchronized with rainy days. A few individuals started calling shortly after sunset and within an hour, choruses were established. On an average the chorus size comprised of about 80–100 individuals spread across the pond. We recorded individuals mostly on semi dry ground due to good accessibility to isolated individuals. Apart from *M. nilphamariensis*, we also observed males of *Minervarya cf. pierrei* to be calling simultaneously.

Individuals of *Microhyla ornata* from the western coast (WC) were recorded from laterite habitats of Konaje, Mangaluru, Karnataka, India (12.8192°N, 74.9316°E, alt 127 m asl; Fig. 1) on 27th July 2016. The laterite plateaus are unique habitats harboring a rich diversity of anurans. Their undulating surfaces and shallow water pools keep anurans concealed while vocalising and provide suitable breeding grounds. The individuals were found vocalising hidden beneath herbaceous plants in these laterite plateaus, at a distance of about 1–5 m from temporary water pools. We recorded individuals that were in chorus on wet laterite rocks. Apart from *M. ornata*, this region is also the type locality of the recently described
congeneric species *Microhyla laterite* (*Seshadri et al., 2016*). Other species vocalising in this region include *Minervarya sahyadris*, *Minervarya caperata* and *Minervarya rufescens*.

## Sound recordings

Sound recordings were made at night (19:30–23:00 h) when *M. nilphamariensis* ($N = 18$) and *M. ornata* (Western Coast; $N = 5$) males vocalized most actively. Phonetically, a typical advertisement call can be described as *trraar-rrataar-trraar-rrataar* in repetition (Call S1 and S2; Video S1 and S2). Calls were recorded on a solid-state digital recorder (Zoom H6 N and Zoom H4N; 44.1 kHz sampling rate; 16 bits sample size) using a unidirectional Sennheiser ME 66 or 67 shotgun microphone and monitored in real-time using Sony MDR ZX110-AP headphones. Microphones were handheld and positioned at a distance of approximately between 50–75 cm from the target male, with settings adjusted before each recording to obtain high signal:noise ratio and the same settings were maintained throughout the recording.

We recorded a minimum of 20 calls from each individual ($N = 23$). Immediately after each recording, males were captured, and SVL (snout to vent length) was measured to the nearest 0.1 mm using dial calipers ($\overline{X} \pm SD = 19.4 \pm 0.7$ mm for *M. nilphamariensis* and $\overline{X} \pm SD = 22.2 \pm 1.2$ mm for *M. ornata*). We also measured body mass to the nearest 0.01 g ($\overline{X} \pm SD = 0.72 \pm 0.05$ g for *M. nilphamariensis* and $\overline{X} \pm SD = 1.5 \pm 0.09$ g for *M. ornata*) using a portable digital balance (Kern CM-60 or American weighing scale$^{\circledR}$). Temperature of the calling site, *i.e.,* wet bulb temperature was recorded using Jennson Delux thermometer or Extech$^{\circledR}$ RH10 portable thermo-hygrometer to the nearest 0.1 °C ($\overline{X} \pm SD = 28.2 \pm 0.8$ for *M. nilphamariensis* and $\overline{X} \pm SD = 28.1 \pm 0.2$ for *M. ornata*). Ambient temperature of the air, *i.e.,* dry bulb temperature was similarly recorded ($\overline{X} \pm SD = 29.5 \pm 0.8$ °C for *M. nilphamariensis* and $\overline{X} \pm SD = 25.6 \pm 0.3$ for *M. ornata*). Individuals were released at their calling site immediately after taking the measurement.

## Acoustic analyses

We analysed 360 calls (20 calls/individual) for *M. nilphamariensis* which we term as advertisement call Type I. In addition, we analysed three other calls of *M. nilphamariensis* that exhibited unusual call characteristics (for both temporal and spectral call properties) and report them as Type II, Type III and Type IV calls. For comparative study, we analysed 100 calls each (20 calls/individual), from the individuals of *M. ornata* recorded from the western coast and from the published acoustic data for *M. ornata* from Sri Lanka (*Wijayathilaka & Meegaskumbura, 2016*).

We used Raven Pro v1.4 (Cornell University, Ithaca, NY, USA) to measure a total of 21 acoustic properties, following *Bee, Suyesh & Biju (2013a)* and *Bee, Suyesh & Biju (2013b)* (Table 1). We measured 14 temporal properties using Raven's waveform display and six spectral properties using Raven's spectrogram function (1024-FFT, Hanning window, 50% overlap, 43.1 Hz resolution). Among temporal properties, we measured call duration, call rise and fall times, pulses/call, pulse rate and periods for the first, middle and last pulses. Within each call, we identified the pulse of maximum amplitude and measured its period, duration, rise time, fall time and pulse 50% onset and offset time. For this, we used Raven's

slice view to accurately measure values of these properties. We also measured inter-call interval which is not a temporal property itself, but is useful for describing call organisation (hence we omitted it from multivariate analysis for species comparision). Among spectral properties, we measured the overall dominant frequency (ODF), which represents the harmonic of the greatest amplitude and dominant frequencies 1 and 2 (DF1 and DF2), which represent two distinct peaks in the call spectrum (*Bee & Gerhardt, 2001*). ODF was measured by selecting the entire call duration using Raven's spectrogram function. DF1 and DF2 were measured by noting the peaks obtained using the spectrogram slice function. Type 1 calls were found to have bimodal frequency spectra, while in Type 2, Type 3 and Type 4 calls we observed a third frequency band. We did not find frequency modulation across any call.

## Statistical analyses
### Descriptive satistics
We computed the mean, standard deviation, minimum and maximum values of the measured acoustic properties for species, *M. nilphamariensis* and *M. ornata* (WC and SL). Since one of our aims was to describe potential sources of variation in calling behaviour in *M. nilphameriensis*, we also calculated Pearson-product moment correlations between call properties and wet and dry bulb temperatures, SVL, mass and body condition. Body condition was calculated as the residuals from a regression of cube root of mass on SVL divided by SVL (*Baker, 1992*). Since these correlation analyses were exploratory and we did not test any particular hypotheses, Bonferroni correction for multiple comparisons was not performed and the significance criterion was set at alpha = 0.05.

We also calculated measures of variability for call properties using coefficients of variation (CV), both among-individuals (CVa) and within-individuals (CVw). CVa was computed as the SD of individual means divided by the average of all individual means ($n = 18$ means). Within-individual coefficient of variation (CVw) was calculated for each male using the mean and SD of the 20 calls recorded for that individual. We report values of CVa, mean and range of CVw and the ratio of CVa:CVw. All these values were calculated using Statistica v7.1 and Microsoft Excel 2010 (Microsoft, Redmond, WA, USA). For all analyses, temperature corrected values of call properties were used to remove effects that may be introduced due to variation in temperature in *M. nilphamariensis* (temperature affects calling behaviour in frogs and this might introduce bias in measuring male-specific variation). This was not done for *M. ornata* individuals from the western coast and Sri Lanka (*Wijayathilaka & Meegaskumbura, 2016*) as they were recorded at a very narrow range of temperature (<0.5 °C).

### Multivariate statistics
We performed multivariate analyses to explore differences in vocal repertoire between *M. nilphamariensis* from Delhi (using Type 1 advertisement calls) and *M. ornata* from the western coast of India and Sri Lanka. We first conducted principal components analysis (PCA) to condense acoustic variation measured across all the individuals into a smaller set of orthogonal components (after standardizing the data set). For this, we used all 20 call properties (excluding inter-call interval) and standardized their values in Statistica. This

 

**Table 1 Description of different properties that are used to describe the male advertisement call of *M. nilphamariensis*.** Properties analyzed are after *Bee, Suyesh & Biju (2013a)* and *Bee, Suyesh & Biju (2013b)*.

| Call properties | Description |
| --- | --- |
| *Overall call properties* | |
| Call duration (ms) | Time between the start of the initial pulse and the end of the final pulse |
| Call rise time (ms) | Time between start of the initial pulse and the point of maximum amplitude (peak) of the call |
| Call fall time (ms) | Time between the point of maximum amplitude and offset of last pulse. |
| Inter-call interval | The time between offset of the last pulse in a call to the beginning of the initial pulse of the next call |
| Pulses per call | Number of pulses in a given call |
| Pulse rate (pulses/s) | Number of pulses minus 1 (N - 1), divided by time between start of the initial pulse and beginning of the last pulse |
| Overall dominant frequency (kHz) | Maximum frequency using Raven's selection spectrum function over the entire call duration |
| Dominant frequency 1 (Hz) | Initial peak frequency using Raven's selection spectrum function over the entire call duration; Initial band in spectrogram view |
| Dominant frequency 2 (Hz) | Second peak frequency using Raven's selection spectrum function over the entire call duration; Second band in spectrogram view |
| *Pulse properties* | |
| Initial pulse period (ms) | Time between start of the initial pulse to start of the second pulse |
| Middle pulse period (ms) | Time between start of the middle pulse to start of the next pulse |
| "N-1" pulse period (ms) | Time between start of the second last pulse to start of last pulse |
| *Maximum amplitude pulse properties* | |
| Overall pulse dominant frequency (kHz) | Peak frequency using Raven's spectrogram slice function over the entire pulse duration |
| Pulse dominant frequency 1 (kHz) | Initial peak frequency using Raven's spectrogram slice function over the entire pulse duration |
| Pulse dominant frequency 2 (kHz) | Second peak frequency using Raven's spectrogram slice function over the entire pulse duration |
| Pulse period (ms) | Time between beginning of pulse to that of the next pulse |
| Pulse duration (ms) | Time between pulse start and end |
| Pulse rise time (ms) | Time between beginning of the pulse and the point where maximum amplitude is attained |
| Pulse 50% rise time (ms) | Time between beginning of the pulse and the point where 50% of the maximum amplitude is attained |
| Pulse fall time (ms) | Time between the point of maximum amplitude and the offset of the pulse |
| Pulse 50% fall time (ms) | Time between the point where the maximum amplitude is half its value (50%) and the offset of the pulse |

generated 20 predictor variables and the corresponding PCA factors, along with a matrix of correlation between the PCA factors and call properties (variables). PCA factors with eigenvalues >1 were then used as input variables for discriminant function analyses (DFA) to obtain discriminant function-based classification of males from all three populations based on the analyzed call properties.

## RESULTS

### Call descriptions of *M. nilphamariensis* from Delhi

Analysis of vocal repertoire of *M. nilphamariensis* revealed four types of calls. Of these, we term the most common call as advertisement call Type I. The other three call types consisted of only three calls samples in total (from two indiviudals). We term these as Type II, Type III and Type IV calls. We describe below the typical advertisement call (Type I), followed by a brief description of Type II, III and IV calls.

### Call properties

Type 1 calls (typical advertisement call) of *M. nilphamariensis* (Fig. 2) did not show any hierarchical call organization such as call bouts or call groups. The calls were sometimes delivered independently or in a series of multiple calls with uniform intervals. The calls typically ranged between 273.6 and 492.2 ms in duration (Table 2). On average, the interval between two calls was $3.6 \pm 4.2$ s ($\overline{X} \pm$ SD). Typically, the calls had pulsatile temporal structure, consisting of 17 pulses per call on average. The amplitude envelope was characterized by a rise time of $174.2 \pm 77.8$ ms, followed by a fall time of $163.9 \pm 53.1$ ms. Pulses were produced at a rate of about $39 \pm 2.5$ pulses/sec (Table 2). The first, middle and N-1 pulse periods were very similar and centered around 25 ms. The call spectrum comprised of two peaks with an overall dominant frequency of $2.8 \pm 0.8$ kHz, while the two peaks (DF1 and DF2) were centered at $1.6 \pm 0.1$ kHz and $3.5 \pm 0.3$ kHz respectively (Table 2, Fig. 2).

Pearson product correlation analyses with SVL, mass, body condition and wet temperature revealed significant correlations for a few measured call properties. Call duration and call rise time were both positively correlated with body condition ($r = 0.58$, $P = 0.01$; $r = 0.64$, $P = 0.00$, respectively). Inter-call interval was also significantly correlated negatively with mass ($r = -0.52$, $P = 0.02$) and body condition ($r = -0.49$, $P = 0.04$). Dominant frequency 1 was significantly correlated negatively with SVL ($r = -0.48$, $P = 0.04$) mass ($r = -0.75$, $P = 0.00$) and body condition ($r = -0.60$; $P = 0.01$). None of the call properties measured were significantly correlated with wet bulb temperature except for first pulse period ($r = -0.53$, $P = 0.02$), although correlations with call rise time and pulses per call were nearly significant ($p = 0.07$; $p = 0.06$, respectively). The measures of central tendency and dispersion for 21 acoustic properties of 360 advertisement calls recorded from 18 individuals is shown in Table 3.

### Pulse properties

The maximum amplitude pulses across individuals were $5.3 \pm 1.3$ ms in duration, while the average pulse period was about $26 \pm 1.8$ ms. Pulses had short rise times at $0.9 \pm 0.2$ ms

Srigyan et al. (2024), *PeerJ*, DOI 10.7717/peerj.16903

Peer⌡

**Table 2 Overview of descriptive statistics of advertisement calls of *Microhyla nilphamariensis* males (N = 18) from Delhi and *M. ornata* from Western Coast (N = 5) and Sri Lanka (N = 5).** Shown here are the means (X), standard deviation (SD) and range of individual means.

| Type of acoustic property | Property | *M. nilphamariensis*, Delhi (N = 18) X̄ | S.D. | Mean range | *M. ornata*, Western Coast (N = 5) X̄ | S.D. | Mean range | *M. ornata*, Sri Lanka (N = 5) X̄ | S.D. | Mean range |
|---|---|---|---|---|---|---|---|---|---|---|
| **Entire call** | | | | | | | | | | |
| Temporal call properties | Call duration (ms) | 393.5 | 57.5 | 274–492.2 | 263.6 | 12.7 | 246.1–281.6 | 292.4 | 22.3 | 262.1–323.7 |
| | Call rise time (ms) | 174.2 | 77.8 | 64.1–304.7 | 147.0 | 25.1 | 117.8–183.5 | 166.2 | 25.4 | 122.8–187.5 |
| | Call fall time (ms) | 163.9 | 53.1 | 74.6–248.5 | 65.4 | 2.4 | 63.2–68.1 | 97.2 | 14.2 | 82.7–118.0 |
| | **# Pulses per call**[*] | **17.0** | **2.4** | **12.5–20** | **11.0** | **2** | **9.6–1.9** | **13.0** | **1** | **12.0–14.0** |
| | Pulse rate (pulses/s) | 39 | 2.5 | 33.8–44 | 38.1 | 1.3 | 36.9–39.8 | 41.6 | 0.9 | 40.4–42.7 |
| Spectral call properties | Overall dominant frequency (kHz) | 2.8 | 0.8 | 1.5–4.1 | 2.7 | 0.1 | 2.5–2.9 | 3.3 | 0.0 | 3.3–3.3 |
| | Overall dominant frequency 1 (kHz) | 1.6 | 0.1 | 1.6–1.7 | 1.3 | 0.0 | 1.2–1.3 | 1.6 | 0.0 | 1.6–1.6 |
| | Overall dominant frequency 2 (kHz) | 3.5 | 0.3 | 3–4.1 | 2.6 | 0.2 | 2.3–2.9 | 3.3 | 0.0 | 3.2–3.3 |
| **Pulse properties** | | | | | | | | | | |
| First, middle and N-1 pulses | First pulse period (ms) | 25.2 | 2.2 | 21.5–31.4 | 25.9 | 1.2 | 24.0–27.1 | 27.7 | 5.0 | 24.7–36.5 |
| | Middle pulse period (ms) | 25.8 | 1.7 | 23.1–29.7 | 26.4 | 0.9 | 25.2–27.5 | 23.9 | 0.2 | 23.6–24.2 |
| | "N-1" pulse period (ms) | 25 | 3.2 | 16.6–28.8 | 26.1 | 2.0 | 23.2–28.6 | 22.6 | 0.4 | 22.0–22.9 |
| Spectral properties maximum pulse | Overall pulse dominant frequency (kHz) | 2.8 | 0.7 | 1.7–3.9 | 2.7 | 0.2 | 2.5–2.9 | 3.3 | 0.0 | 3.3–3.3 |
| | Pulse dominant frequency 1 (kHz) | 1.6 | 0.1 | 1.6–1.7 | 1.3 | 0.0 | 1.2–1.3 | 1.6 | 0.0 | 1.6–1.7 |
| | Pulse dominant frequency 2 (kHz) | 3.6 | 0.3 | 3–4 | 2.7 | 0.2 | 2.6–3.0 | 3.3 | 0.0 | 3.3–3.3 |
| Temporal properties maximum pulse | Pulse period (s) | 25.9 | 1.8 | 22.4–29.9 | 26.4 | 0.9 | 25.1–27.3 | 24.2 | 0.4 | 23.6–24.5 |
| | Pulse duration (ms) | 5.3 | 1.3 | 3.5–8.5 | 5.1 | 1.1 | 3.6–6.5 | 5.4 | 0.5 | 4.5–5.8 |
| | Pulse rise time (ms) | 0.9 | 0.2 | 0.7–1.3 | 1.2 | 0.2 | 1.1–1.4 | 1.3 | 0.0 | 1.2–1.3 |

Srigyan et al. (2024), *PeerJ*, DOI 10.7717/peerj.16903

**Table 2** (*continued*)

| Type of acoustic property | Property | *M. nilphamariensis*, Delhi (N = 18) | | | *M. ornata*, Western Coast (N = 5) | | | *M. ornata*, Sri Lanka (N = 5) | | |
|---|---|---|---|---|---|---|---|---|---|---|
| | | $\overline{X}$ | S.D. | Mean range | $\overline{X}$ | S.D. | Mean range | $\overline{X}$ | S.D. | Mean range |
| **Entire call** | | | | | | | | | | |
| | Pulse 50% rise time (ms) | 0.7 | 0.1 | 0.5–0.9 | 0.9 | 0.1 | 0.8–1.0 | 0.7 | 0.0 | 0.6–0.7 |
| | Pulse fall time (ms) | 4.3 | 1.4 | 2.4–7.8 | 4.1 | 1.0 | 2.5–5.4 | 4.2 | 0.5 | 3.3–4.5 |
| | Pulse 50% fall time (ms) | 3.7 | 1.3 | 1.9–6.3 | 3.3 | 1.0 | 1.7–4.6 | 3.5 | 0.5 | 2.7–3.9 |

**Notes.**

*For pulses per call (highlighted in bold), the values reported in the columns headed X and SD are the median and interquartile range.

Srigyan et al. (2024), *PeerJ*, DOI 10.7717/peerj.16903

**Table 3 Correlation analysis and coefficients of variation.** (A) Pearson-product moment correlations (*r*) between *Microhyla nilphamarensis* (N = 18) call properties with snout-to-vent length (SVL), mass, body condition and temperature. (B) Coefficients of variation computed among (CVa) and within males (CVw) and their ratios.

| | | A. Correlation analysis | | | | | | | | B. Coefficients of variation | | |
|---|---|---|---|---|---|---|---|---|---|---|---|---|
| Type of acoustic property | Property | SVL | | Mass | | Body condition | | Wet bulb temperature (°C) | | CVa | CVw mean (range) | CVa:CVw |
| Call properties | | r | P | r | P | r | P | r | P | | | |
| Temporal call properties | Call duration (ms) | −0.27 | 0.29 | 0.37 | 0.14 | **0.58** | **0.01**[**] | 0.40 | 0.10 | 14.6 | 10.0 (3.44–18.81) | 1.46 |
| | Call rise time (ms) | −0.26 | 0.30 | 0.43 | 0.07[*] | **0.64** | **0.00**[**] | 0.43 | 0.07[*] | 44.7 | 26.05 (7.61–73.73) | 1.72 |
| | Call fall time (ms) | 0.15 | 0.54 | −0.20 | 0.43 | −0.31 | 0.21 | −0.35 | 0.16 | 32.4 | 32.1 (14.1–48.0) | 1.01 |
| | Intercall interval (s) | −0.23 | 0.37 | **−0.53** | **0.02**[**] | **−0.50** | **0.04**[**] | −0.09 | 0.71 | 116.7 | 150.6 (6.4–327.0) | 0.77 |
| | Pulses per call | −0.29 | 0.25 | 0.19 | 0.45 | 0.39 | 0.11 | 0.45 | 0.06[*] | 13.2 | 9.9 (3.5–19.2) | 1.33 |
| | Pulse rate (pulses/s) | −0.11 | 0.68 | −0.40 | 0.10 | −0.40 | 0.10 | 0.28 | 0.25 | 6.5 | 2.7 (1.1–7.4) | 2.39 |
| Spectral call properties | Overall dominant frequency (peak) (kHz) | −0.05 | 0.86 | 0.06 | 0.82 | 0.09 | 0.71 | 0.04 | 0.89 | 28.8 | 16.2 (0.0–43.9) | 1.78 |
| | Overall dominant frequency 1 (kHz) | **−0.48** | **0.04**[**] | **−0.75** | **0.00**[**] | **−0.60** | **0.01**[**] | 0 | 1.0 | 3.4 | 0.7 (0.0–1.6) | 4.68 |
| | Overall dominant frequency 2 (kHz) | −0.18 | 0.48 | 0.09 | 0.74 | 0.20 | 0.42 | −0.06 | 0.80 | 7.2 | 2.2 (0.0–8.3) | 3.33 |
| Pulse properties | | | | | | | | | | | | |
| First, middle and N-1 pulses | First pulse period (ms) | −0.07 | 0.79 | 0.18 | 0.48 | 0.24 | 0.34 | **−0.53** | **0.02**[**] | 8.9 | 15.8 (4.5–50.2) | 0.56 |
| | Middle pulse period (ms) | 0.11 | 0.65 | 0.43 | 0.07[*] | 0.43 | 0.07[*] | −0.13 | 0.61 | 6.6 | 3.5 (1.9–14.4) | 1.88 |
| | "N-1" pulse period (ms) | 0.12 | 0.64 | 0.08 | 0.76 | 0.01 | 0.96 | −0.26 | 0.30 | 12.9 | 13.4 (6.1–27.5) | 0.96 |

Srigyan et al. (2024), *PeerJ*, DOI 10.7717/peerj.16903

**Table 3** (*continued*)

| Type of acoustic property | Property | SVL | | Mass | | Body condition | | Wet bulb temperature (°C) | | CVa | CVw mean (range) | CVa:CVw |
|---|---|---|---|---|---|---|---|---|---|---|---|---|
| | | | | | | **A. Correlation analysis** | | | | | **B. Coefficients of variation** | |
| Spectral properties maximum pulse | Overall pulse dominant frequency (kHz) | −0.14 | 0.58 | 0.20 | 0.43 | 0.30 | 0.22 | 0.24 | 0.33 | 26.1 | 18.4 (0.0–43.1) | 1.42 |
| | Pulse dominant frequency 1 (kHz) | **−0.62** | **0.00**[**] | **−0.66** | **0.00**[**] | −0.42 | 0.09[*] | 0.21 | 0.41 | 3.3 | 2.2 (0.0–7.5) | 1.49 |
| | Pulse dominant frequency 2 (kHz) | −0.24 | 0.33 | −0.084 | 0.74 | 0.04 | 0.87 | −0.12 | 0.63 | 8.0 | 3.7 (0.9–7.6) | 2.17 |
| Temporal properties maximum pulse | Pulse period (s) | 0.18 | 0.49 | 0.30 | 0.23 | 0.24 | 0.33 | −0.20 | 0.43 | 7.1 | 4.6 (1.9–26.9) | 1.52 |
| | Pulse duration (ms) | 0.09 | 0.73 | −0.01 | 0.97 | −0.06 | 0.82 | −0.10 | 0.69 | 25.4 | 12.6 (3.2–22.3) | 2.02 |
| | Pulse rise time (ms) | −0.27 | 0.27 | 0.24 | 0.33 | 0.43 | 0.07[*] | −0.18 | 0.47 | 16.4 | 14.9 (1.7–42.8) | 1.10 |
| | Pulse 50% rise time (ms) | 0.25 | 0.31 | **0.50** | **0.03**[**] | 0.43 | 0.08[*] | −0.30 | 0.23 | 15.2 | 14.0 (1.7–26.3) | 1.09 |
| | Pulse fall time (ms) | 0.12 | 0.65 | −0.04 | 0.89 | −0.10 | 0.68 | −0.08 | 0.77 | 32.4 | 16.9 (5.2–38.7) | 1.92 |
| | Pulse 50% fall time (ms) | 0.06 | 0.82 | −0.03 | 0.91 | −0.07 | 0.80 | −0.08 | 0.76 | 35.1 | 19.4 (8.9–37.1) | 1.81 |

**Notes.**

[*]For pulses/call, median and inter-quartile ranges are reported here instead of means and S.D. Sri Lanka; quartiles: q1 =12.5, q2 = 13, q3 = 13.5; Western Coast; quartiles: q1 = 9.5, q2 = 11, q3 = 11.5; New Delhi: q1 =15, q2 = 16.5, q3 = 18.75.

[**]Bold type indicates statistically significant correlations ($P < 0.05$, p-values marked with (**)), while marginally significant correlations are marked with an asterisk (*) for $0.05 < p < 0.09$).

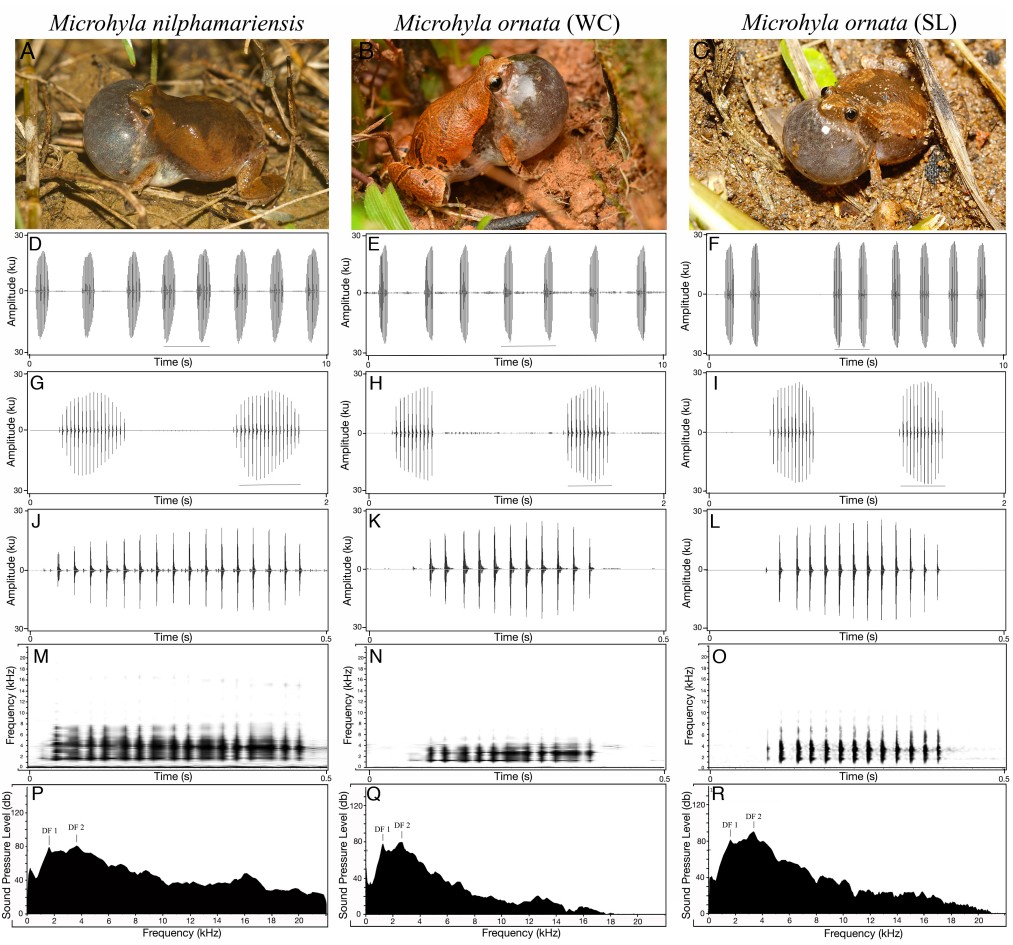

**Figure 2  Vocal repertoire of *Microhyla nilphamariensis* and *Microhyla ornata* (WC and SL).**  (A–C) Vocalising animal. (D–F) Waveform of the call (10 s sequence). (G–I) Waveform of call (2 s sequence) showing two calls. (J–L) Waveform of a single call (0.5 s). (M–O) Spectrogram of single call (0.5 s). (P–R) Power spectrum of a call showing two dominant frequency peaks. (Figure for *M. nilphamariensis* is based on Type 1 calls).

with 50% of maximum amplitude attained in $0.7 \pm 0.1$ ms. The average pulse fall time at $4.3 \pm 1.4$ ms was nearly five times longer than the pulse rise time. The pulse decreased to 50% of its maximum amplitude (pulse 50% fall time) nearly $3.7 \pm 1.3$ ms before its end. Spectral characteristics of pulses were similar to those of calls, with a pulse overall dominant frequency (ODF) of $2.8 \pm 0.7$ kHz, while pulse dominant frequencies DF1 and DF2 were $1.6 \pm 0.1$ kHz and $3.6 \pm 0.3$ kHz, respectively. Significant correlation was obtained for only two pulse properties, *i.e.,* pulse dominant frequency 1, which was negatively correlated with SVL ($r = -0.6201$, $P = .006$) and mass ($r = -0.6553$, $P = 0.003$); and for pulse 50% rise time, which was positively correlated with mass ($r = 0.5008$, $P = 0.034$).

## Patterns of variability in properties

Out of the 21 total call properties measured, 15 properties exhibited relatively greater variation among individuals than within individuals (CVa:CVw >1). Overall, dominant

frequency 1 and 2 were found to have the highest CVa:CVw ratio, followed by pulse rate (Table 3). Such properties have implications for individual discrimination (*Bee & Gerhardt, 2001*; *Bee et al., 2001*). Of the remaining six, call fall time, N-1 pulse period, pulse rise time and 50% pulse rise time varied similarly both among and within individuals ($0.96 \leq$ CVa:CVw $\leq 1.1$), while first pulse period and inter-call interval were more variable within individuals than among (CVa:CVw ratios 0.55 and 0.66, respectively).

Within individuals, the degree of variability in call properties has led to their classification as "static" or "dynamic" (*Gerhardt, 1991*). Static properties often include properties that are constrained physically, are important for species recognition, and show less within-individual variability (typically ≤5%) than dynamic properties (*e.g.*, ≥12%) (*Bee, Suyesh & Biju, 2013a*). Within individuals of *M. nilphamariensis*, seven properties had CVw <5%, out of which four were spectral (call and pulse dominant frequencies 1 and 2) and three were temporal (pulse rate, middle pulse period and pulse period of the maximum amplitude pulse). Following this criterion, we categorize these properties as "static". Most properties having within-individual variation above 12% were temporal, although overall dominant frequency for both calls and pulses had CVw >12%. This can be explained since the values of ODF measured in each call shuffled between the two spectral peaks and thus the CV captures this variability. The highest values were seen for Inter-call Interval and Call rise time and would be assigned "dynamic" on the variability spectrum. Remaining properties had ≤5% CVw ≤12% and were "intermediate" (Table 3).

### Different call types observed in *M. nilphamariensis*

Apart from the 20 calls measured for each of the 18 *M. nilphamariensis* individuals, we observed three calls from two individuals that differed significantly from the rest of the data in this population, labelled as Type II, III and IV (Fig. 3). We observed two calls (labelled as Type II and Type III) in another individual that exhibited unusual characteristics which notably included three frequency peaks as well as a lower-than-average pulse rate (28.3 and 32.4 pulses/s, respectively) for the individual, which made them different from Type I (Table 4). Thus, these calls (Type III and Type IV), which exhibit trimodal spectra and are unique in the otherwise bimodal frequency spectrum observed for *M. nilphamariensis* (Table 4, Fig. 3). Type III and Type IV calls differed from each other in both temporal as well as spectral properies (Table 4). For example, call duration (426.5 ms *vs.* 96.7 ms), call rise time (201.8 ms *vs* 30.4 ms), call fall time (90.2 ms *vs.* 33.8 ms), overall dominant frequency (3.4 kHz *vs* 1.6 kHz) for Type II and Type III call, respectively (Table 4). Further, in a call (labelled as Type IV) of one individual, values of all temporal properties were different compared to Type I, Type II and Type III calls (Table 4). For example, call duration (15.1 ms *vs.* 393.5 ms *vs* 426.5 ms *vs.* 96.7 ms), call rise time (9.1 ms *vs* 174.2 ms *vs* 201.8 ms *vs* 30.4 ms), call fall time (6.0 ms *vs* 163.9 ms *vs* 90.2 ms *vs.* 33.8 ms), pulse rate (117.0 pulses/s *vs* 39.0 pulses/s *vs* 28.3 and 32.4 pulses/s) for Type IV, Type I, Type II and Type III calls, respectively (Table 4). Type II calls also exibited trimodal frequency spectra like Type II and Type III calls, unlike what was seen in the Type 1 calls spectral call properties of Type II and Type I calls were found to be similar (Table 4, Fig. 3).

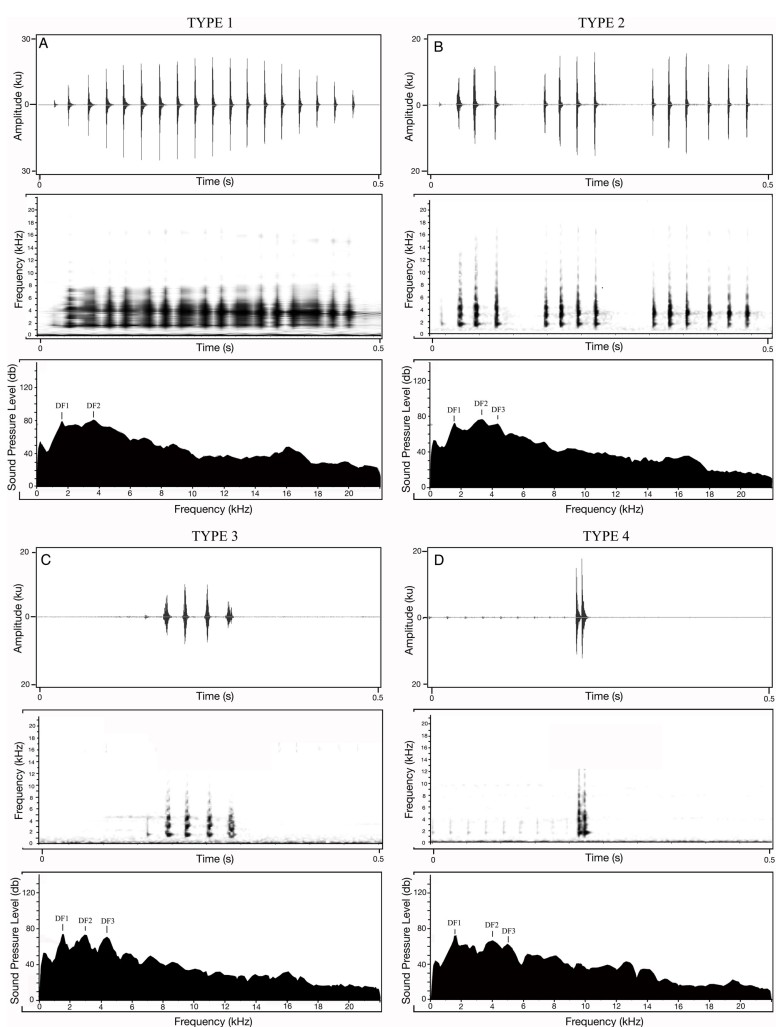

**Figure 3** **Different call types of *Microhyla nilphamariensis*.** (A) Waveform of Type I call (0.5 s); spectrogram of type I call (0.5 s); power spectrum of type I call showing two dominant frequency peaks. (B) Waveform of type II call (0.5 s); spectrogram of type II call (0.5 s); power spectrum of type II call showing three dominant frequency peaks. (C) Waveform of type III call (0.5 s); spectrogram of type III call (0.5 s); power spectrum of type III call showing three dominant frequency peaks. (D) Waveform of Type IV call (0.5 s); spectrogram of Type IV call (0.5 s); power spectrum of type IV call showing two dominant frequency peaks.

## Call descriptions of *M. ornata*
### *Microhyla ornata*, western coast

*Call properties.* The measures of central tendency and dispersion for 20 acoustic properties of 100 advertisement calls recorded from five *M. ornata* WC individuals are shown in Table 2. The advertisement call (Fig. 2) duration ranged between 246–281 ms, and a typical call consisted of $11 \pm 2$ pulses on average. Call rise and fall times were measured to be $147 \pm 25.1$ ms and $65.4 \pm 2.4$ ms, while pulse rate was found to be $38.1 \pm 1.3$ pulses/s. The first, middle and N-1 pulse periods were very similar, around 26 ms on average. Among the spectral properties, overall call dominant frequency was found to be $2.7 \pm 0.1$ kHz, while

**Table 4 Description of rare call types.** Type 2, Type 3 and Type 4 in the male advertisement call of *M. nilphamariensis*, along with mean values of call properties for Type I call shown for comparison.

| Type of acoustic property | Property | Type II | Type III | Type IV | Type 1 $\overline{X}$ |
|---|---|---|---|---|---|
| Entire call | | | | | |
| Temporal call properties | Call duration (ms) | 426.5 | 96.7 | 15.1 | 393.5 |
| | Call rise time (ms) | 201.8 | 30.4 | 9.1 | 174.2 |
| | Call fall time (ms) | 90.2 | 33.8 | 6.0 | 163.9 |
| | **# Pulses per call[*]** | **13.0** | **4.0** | **2.0** | **17.0** |
| | Pulse rate (pulses/s) | 28.3 | 32.4 | 117.0 | 39 |
| Spectral call properties | Overall dominant frequency (kHz) | 3.4 | 1.6 | 1.7 | 2.8 |
| | Dominant frequency 1 (kHz) | 1.6 | 1.6 | 1.6 | 1.6 |
| | Dominant frequency 2 (kHz) | 3.3 | 3.0 | 3.7 | 3.5 |
| | Dominant frequency 3 (kHz) | 4.3 | **4.3** | 5.0 | **NA** |
| Pulse properties | | | | | |
| First, middle and N-1 pulses | First pulse period (ms) | 23.6 | 28.3 | 8.5 | 25.2 |
| | Middle pulse period (ms) | 85.1 | 31.7 | NA | 25.8 |
| | "N-1" pulse period (ms) | 26.8 | 31.7 | 8.5 | 25 |
| Spectral properties maximum pulse | Overall pulse dominant frequency (kHz) | 3.4 | 3.0 | 1.7 | 2.8 |
| | Pulse dominant frequency 1 (kHz) | 1.6 | 1.6 | 1.7 | 1.6 |
| | Pulse dominant frequency 2 (kHz) | 3.4 | 3.0 | 4.0 | 3.6 |
| | Pulse dominant frequency 3 (kHz) | 4.3 | **4.3** | 5.0 | NA |
| Temporal properties maximum pulse | Pulse period (s) | 85.1 | 32.5 | 8.5 | 25.9 |
| | Pulse duration (ms) | 3.1 | 4.3 | 6.0 | 5.3 |
| | Pulse rise time (ms) | 1.0 | 2.0 | 0.5 | 0.9 |
| | Pulse 50% rise time (ms) | 0.5 | 1.3 | 0.5 | 0.7 |
| | Pulse fall time (ms) | 2.1 | 2.3 | 5.4 | 4.3 |
| | Pulse 50% fall time (ms) | 1.3 | 0.9 | 4.7 | 3.7 |

**Notes.**
*For pulses per call (highlighted in bold), the values shown under the column headed Type I (X), is median.

call DF1 and DF2 were measured to be 1.3 kHz and 2.3 kHz respectively, thus markedly different from *M. ornata* (SL) described below (Table 2, Fig. 2).

## Pulse properties

For the maximum amplitude pulse, values of temporal properties were very similar to those of *M. ornata* (SL). The average pulse duration was $5.1 \pm 1.1$ ms, while the pulse period was $26.4 \pm 0.9$ ms. The average pulse rise and fall time were measured to be $1.2 \pm 0.2$ ms and $4.1 \pm 1.2$ ms. Similarly, pulse 50% rise and fall time were found to be $0.9 \pm 0.1$ ms and $3.3 \pm 1.0$ ms. Values of pulse spectral properties were almost identical to call spectral properties as seen in all three populations, *i.e.,* overall pulse DF, pulse DF1 and pulse DF2 were found to be $2.7 \pm 0.2$, 1.3 and $2.7 \pm 0.2$ kHz, respectively (Table 2).

## *Microhyla ornata,* Sri Lanka
### Call properties

Similar to *M. ornata* (WC), the descriptive statistics for the 20 measured call properties across 100 calls from five *M. ornata* individuals from Sri Lanka are listed in Table 2. A typical

advertisement call of *M. ornata* (Fig. 2) had a pulsatile temporal structure consisting of 13 pulses on average. Advertisement calls typically ranged between 262–323 ms in duration (Table 2). On average, the call rise time was 166.2 ± 25.4 ms while call fall time was 97.2 ± 14.2 ms. The typical pulse rate was 41.6 ± 0.9 pulses/s, *i.e.,* slightly higher than that of *M. ornata* (WC). The first, middle and N-1 pulse periods were averaged roughly between 22–28 ms. The call spectrum consisted of an overall dominant frequency of 3.3 kHz, with two frequency peaks corresponding to dominant frequency 1 & 2 at 1.6 kHz and 3.3 kHz respectively (Table 2, Fig. 2).

## Pulse properties

For the maximum amplitude pulse, the average pulse duration was 5.4 ± 0.5 ms, while pulse period was longer, at about 24.2 ± 0.4 ms. Pulses had an average rise time of 1.3 ms, while pulse fall time was nearly three times longer, at 4.2 ms. The pulse 50% rise time was short, at 0.7 ms, while pulse 50% fall time was much longer, at 3.5 ± 0.5 ms. The pulse spectral properties measured were identical to the calls, *i.e.,* pulse ODF, pulse DF 1 and pulse DF 2 were 3.3, 1.6 and 3.3 kHz respectively (Table 2).

## Overall patterns of acoustic differences
### Vocal repertoire of *Microhyla nilphamariensis* and *Microhyla ornata*

When summarizing the vocal repertoire of the three study populations involving two species, a few qualitative patterns emerge. Across temporal properties, average values of call properties for *M. nilphamariensis* were higher than those for both populations of *M. ornata* (*M. nilphamariensis vs.* SL and WC), while pulse temporal property values were very similar across the three groups (Table 2). For example, call duration (393.5 ± 57 ms *vs.* 292.4 ± 22.3 ms (SL) and 263.6 ± 12.7 ms (WC)), call rise time (174.2 ± 77.8 ms *vs* 166.2 ± 25.4 ms (SL) and 147 ± 25.1 ms (WC)), call fall time (163.9 ± 53.1 ms *vs.* 97.2 ± 14.2 (SL) and 65.4 ± 2.4 ms (WC)), pulses/call (16.5 ± 2.4 *vs.* 13 ± 1 (SL) and 11 ± 2 (WC)) were markedly higher in *M. nilphamariensis*, and the overall higher values of SD indicate that variability within *M. nilphamariensis* was also greater (sampling effect). For the first, middle and N-1 pulses, average pulse period values were similar across the two species and ranged from about 23–27 ms (Table 2). The highest similarity across the groups was seen in temporal values of the maximum amplitude pulse, *i.e.,* pulse duration, pulse period, pulse rise and fall time, pulse 50% rise and fall time, wherein many values were almost identical for both species (Table 2). Values of pulse rate were also found to be similar across the three (39 ± 2.9 pulses/s *vs.* 41.6 ± 0.9 pulses/s (SL) and 38.1 ±1.3 pulses/s (WC)), with the highest rate seen in *M. ornata* (SL) (Table 2).

Among the spectral properties, patterns of differences were inconsistent across the three groups *i.e.,* properties of *M. nilphamariensis* were not always higher in magnitude compared to the two *M. ornata* groups as in call properties summarized above. For example overall call dominant frequency for *M. nilphamariensis* was found to be about 2.8 kHz, while the values for *M. ornata* (SL) and *M. ornata* (WC) were approx. 3.3 kHz and 2.7 kHz respectively. Hence, while the two *M. ornata* populations would be expected to have similar values of overall dominant frequency (ODF), *M. nilphamariensis* and *M. ornata* (WC) are

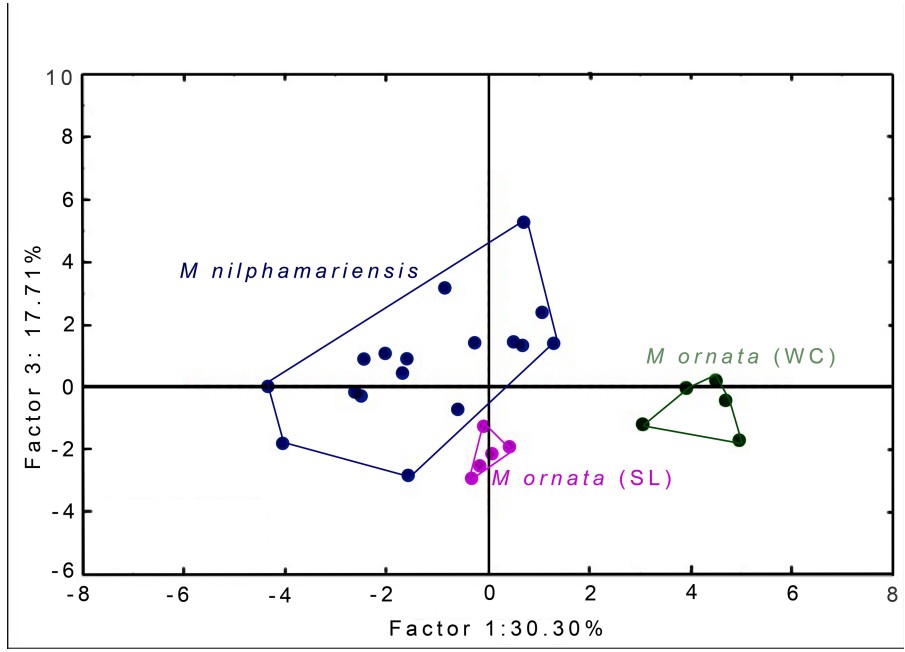

**Figure 4** PCA Plots. Segregation of *M. nilphamariensis* and populations of *M. ornata* (SL & WC) in different factor planes.

instead more similar to each other. However, for call DF1 and call DF2 peaks, where *M. nilphamariensis* & *M. ornata* (SL) are similar in the average values of these properties *i.e.,* DF1 = 1.6 and 1.3 kHz; DF2 = 3.5 and 3.3 kHz respectively for *M. nilphamariensis* and *M. ornata* (SL). This pattern is replicated in pulse spectral properties, as would be expected (Table 2). Hence, *M. nilphamariensis* is similar to M. ornata (WC) in overall call dominant frequency but similar to M. ornate (SL) in call DF1 and DF2 peaks.

**Multivariate statistical comparison (*M. nilphamariensis* and *M. ornata*)**

PCA was performed using all 20 measured call properties of Type 1 call to elucidate the pattern of variation among the two species based on vocal repertoire, followed by discriminant function analysis to investigate the degree of distinctiveness (based on principal component factors) between them.

PCA generated 20 factors, their eigenvalues and factor loading scores, *i.e.,* correlation with call properties. The first five factors had eigenvalue >1 and together accounted for 86.05% of total variation (Table 5). Factor-variable correlation scores (threshold of $r \geq 0.6$) revealed that 18 out of 20 call properties (variables) were highly correlated to one of the first five PCA factors. From the PCA factor planes, we observed that *M. nilphamariensis* individuals are widely dispersed, whereas *M. ornata* from Sri Lanka and western coast form distinct clusters (Fig. 4).

PC factor 1, which explains 30.30% of the total variation was highly correlated negatively with seven acoustic properties, i.e, call duration ($r = -0.61$), call fall time ($r = -0.70$), pulses/call ($r = -0.74$), call DF1 ($r = -0.78$) and DF2 ($r = -0.83$), as well as pulse

**Table 5 Results from a principal components analysis.** Shown here are the first five factors (out of 20) that had eigenvalues >1 and together accounted for 86.06% of total variation. Factor-variable correlation scores (threshold of $r \leq 0.6$, marked in bold (*) shown here for the first five PC factors revealed that 18 out of 20 call properties (variables) were highly correlated to one of these five PCA factors. (All 20 factors and their eigenvalues shown as supplementary information in Tables S1 & S2).

| Type of acoustic property | Property | Factor 1 | Factor 2 | Factor 3 | Factor 4 | Factor 5 |
|---|---|---|---|---|---|---|
| Call Properties | | | | | | |
| Temporal call properties | Call duration (s) | **−0.61***  | 0.32 | 0.48 | −0.49 | −0.11 |
| | Call rise time (s) | 0.05 | **0.65*** | 0.19 | **−0.66*** | −0.02 |
| | Call fall time (s) | **−0.70*** | −0.38 | 0.37 | 0.30 | −0.01 |
| | #Pulses per call | **−0.74*** | 0.34 | 0.24 | −0.45 | −0.17 |
| | Pulse rate (pulses/s) | −0.42 | 0.11 | **−0.88*** | −0.04 | −0.09 |
| Spectral call properties | Overall dominant frequency (peak) (kHz) | 0.11 | **0.68*** | −0.16 | 0.03 | 0.26 |
| | Overall dominant frequency 1 (peak) (kHz) | **−0.78*** | 0.36 | 0.05 | 0.40 | 0.07 |
| | Overall dominant frequency 2 (peak) (kHz) | **−0.83*** | 0.27 | 0.31 | 0.01 | 0.14 |
| Pulse properties | | | | | | |
| First, middle and N-1 pulses | First pulse period (s) | 0.19 | −0.01 | 0.30 | 0.14 | **0.76*** |
| | Middle pulse period (s) | 0.39 | −0.08 | **0.86*** | −0.09 | −0.01 |
| | "N-1" pulse period (s) | 0.41 | −0.16 | **0.63*** | 0.39 | −0.01 |
| Spectral properties maximum pulse | Overall pulse dominant frequency (kHz) | 0.23 | **0.85*** | 0.03 | −0.09 | 0.32 |
| | Pulse dominant frequency 1 (kHz) | **−0.76*** | 0.42 | 0.12 | 0.36 | 0.05 |
| | Pulse dominant frequency 2 (kHz) | **−0.86*** | 0.12 | 0.28 | −0.03 | 0.12 |
| Temporal properties maximum pulse | Pulse period (s) | 0.37 | −0.14 | **0.85*** | −0.01 | −0.01 |
| | Pulse duration (ms) | −0.49 | **−0.71*** | −0.13 | −0.24 | 0.33 |
| | Pulse rise time (ms) | 0.48 | 0.30 | −0.29 | −0.16 | 0.56 |
| | Pulse 50% rise time (ms) | 0.51 | −0.52 | 0.07 | −0.44 | −0.02 |
| | Pulse fall time (ms) | −0.49 | **−0.76*** | −0.09 | −0.23 | 0.24 |
| | Pulse 50% fall time (ms) | −0.56 | **−0.73*** | −0.04 | −0.20 | 0.30 |
| | **Eigen value** | 6.06 | 4.42 | 3.54 | 1.81 | 1.38 |
| | **Variance (%)** | 30.30 | 22.09 | 17.71 | 9.04 | 6.92 |
| | **Cumulative (%) of variance** | 30.30 | 52.39 | 70.10 | 79.14 | 86.06 |

DF1 ($r = -0.76$) and pulse DF2 ($r = -0.85$). PC factor 2, which explained 22.08% of the variation was highly correlated positively with call rise time ($r = 0.65$), overall call dominant frequency (ODF) ($r = 0.67$), pulse ODF ($r = 0.85$) and negatively correlated with pulse duration ($r = -0.71$), pulse fall time ($r = -0.76$) and pulse 50% fall time ($r = -0.73$). Factor 3 was correlated negatively with pulse rate ($r = -0.88$) and positively with middle pulse period ($r = 0.86$), N-1 pulse period ($r = 0.63$) and pulse period for the maximum amplitude pulse ($r = 0.85$). Factor 4 was negatively correlated with call rise time while factor 5 was positively correlated with first pulse period (Table 5). Values of factor-variable correlations of all 20 PC factors as well as their eigenvalues are given in Tables S1 and S2.

**Table 6 Classification matrix.** Obtained from Discriminant Function Analyses (DFA) run on PCA scores (first five factors with eigenvalue > 1) showing the classification of all the males of *Microhyla nilphamariensis* (Delhi) and *Microhyla ornata* (Western Coast and Sri Lanka) in the study into groups defined by locality.

| Group | Percent correct | *Delhi* $p = .64286$ | *Western coast* $p = .17857$ | *Sri Lanka* $p = .17857$ |
|---|---|---|---|---|
| *Delhi* | 100 | 18 | 0 | 0 |
| *Western Coast* | 100 | 0 | 5 | 0 |
| *Sri Lanka* | 100 | 0 | 0 | 5 |
| Total | 100 | 18 | 5 | 5 |

**Table 7 Discriminant function roots.** Standardized canonical discriminant function coefficients showing the relative importance of the first five PCA factor scores in the composition of the discriminant functions (significant values are highlighted in bold). Two discriminant roots were generated, placing all individuals into their respective populations, resulting in 100% classification.

| | Root 1 | Root 2 |
|---|---|---|
| PCA_Factor1 | **2.32** | 0.23 |
| PCA_Factor2 | **−1.31** | 0.61 |
| PCA_Factor3 | **−1.63** | −0.79 |
| PCA_Factor4 | −0.93 | 0.46 |
| PCA_Factor5 | 0.78 | **0.96** |
| Eigenvalue | 24.66 | 2.20 |
| **Variance (%)** | 91.80 | 8.20 |
| **Cumulative (%) of variance** | 91.80 | 100 |

When raw factor scores from PCA were used as input variables for DFA, we found that all males were correctly assigned to their source locality (Table 6, Table S3). Two discriminant functions (roots) were generated, placing all individuals into their respective populations, resulting in 100% classification (Tables 6 and 7). The first root had eigenvalue 24.66, while that of the second root was only 2.2, thus the first root explains most of the discrimination obtained. When taking values of raw coefficients >1.0, Root 1 was correlated with PC factor 1 ($r = 4.03$), followed by factor 3 ($r = −1.97$) and factor 2 ($r = −1.38$), while Root 2 was highly correlated only with factor 5 ($r = 1.13$). Since Root 1 had maximum correlation with factor 1 and factor 3, PCA factor planes (shown above, Fig. 4) have been plotted using these two factors (F1 x F3). Values of the raw as well as standardised canonical roots (discriminant functions) along with their correlation with PC Factors (variables) are given in Table 7.

When PCA factor loadings and DFA factor structure are analysed jointly, candidates for call properties that have maximally contributed to distinction between the three groups can be identified. The first canonical DFA root which accounted for 91.8% of the cumulative proportion, was most highly correlated with PC factor 1, PC factor 2 and PC factor 3. PC factor 1 loaded most heavily on call and pulse dominant frequencies 1 and 2, followed by pulses/call, call fall time and call duration. Factor 2 loaded most heavily on pulse ODF followed by pulse temporal properties (duration, fall time, 50% fall time), Call ODF and call

rise time. Factor 3 loaded most on pulse rate, followed by pulse periods of the middle, the maximum amplitude pulse, and the n-1 pulse. The second canonical root was correlated with PCA factor 5, which loaded only on first pulse period, Thus, out of all the highly correlated properties, it appears that spectral properties such as call and pulse dominant frequencies and temporal properties such as pulse rate, pulses/call, call duration, pulse duration *etc.* have contributed heavily towards statistically discriminating among the three groups involving these two species.

## DISCUSSION

In recent years, even though there has been a surge of studies reporting new *Microhyla* species from the Indian subcontinent (*Garg & Biju, 2019*; *Biju et al., 2019*), their acoustic descriptions have not kept up with this pace despite being crucial for informing conservation related actions. Urban areas are particularly under-sampled regions, resulting in limited reports of anurans from these areas (*Vineeth et al., 2018*) despite the threats posed by elevated anthropogenic pressures on wildlife and their effects on anuran communication (*Rabin & Greene, 2002*; *Warren et al., 2006*; *Bee & Swanson, 2007*). Hence, one of our primary aims was to generate a comprehensive account of the vocal repertoire of *M. nilphamariensis* in Delhi. To our knowledge, this is the first extensive report of calling behaviour of this species. Secondly, we analyse the call properties of *M. ornata* from the western coast of India and Sri Lanka (*Wijayathilaka & Meegaskumbura, 2016*) and compare the vocal behaviour of both the species. As expected calls of *M. nilphamariensis* from Delhi can be differentiated from *M. ornata* from both western coast as well as Sri Lanka, while *M. ornata* populations also seem to be vocally distinct.

### Vocal repertoire of *M. nilphamariensis* and *M. ornata*

We sampled a large number of representative individuals and characterised spectral and temporal properties of their calls. A large sample size is necessary to account for variation in calling behaviour and standardise call structure. We describe one main advertisement call as well as three 'rare' call types (Fig. 3). Studies that include acoustic descriptions of *Microhyla* elsewhere, including that of *M. nilphamariensis* and/or *M. ornata* (India) have sampled fewer individuals, measure only a small number of call properties or have reported only one call type (*Kuramoto & Joshy, 2006*; *Hasan et al., 2015*; *Garg & Biju, 2019*), leaving significant intra-specific acoustic variation undocumented. Analysis of call characters for both *M. nilphamariensis* and the two populations of *M. ornata* revealed pulsatile call structure (Fig. 2), which has been reported across the genus (see *Wijayathilaka & Meegaskumbura, 2016*). For *M. ornata*, call properties overall were found to be similar to that reported for *M. ornata* in Sri Lanka and in India (*Wijayathilaka & Meegaskumbura, 2016*; *Garg & Biju, 2019*).

### Patterns and sources of variability (*M. nilphamariensis*)

The vocal repertoire of *M. nilphamariensis* comprises of a dominant call type, although we noted presence of a few calls that differed in some temporal and spectral properties. Production of diverse call types is common across frog species (*Narins & Capranica,*

1978; *Narins, Lewis & McClelland, 2000*; *Christensen-Dalsgaard, Ludwig & Narins, 2002*; *Feng, Narins & Xu, 2002*) and combining different notes to create complex calls has also been reported frequently (*Rand & Ryan, 1981*; *Wells & Schwartz, 1984*). Notably, three of these calls (Type II, Type III and Type IV) was observed to have trimodal frequency spectra (Fig. 3). Different dominant frequencies are likely produced by differential filtering through resonating structures during calling and are suggested to have a role in anuran signal perception, since both their hearing organs (*i.e.,* amphibian papilla and the basilar papilla) are tuned to different frequency ranges (*Fritzsch & Wake, 1988*; *Ryan & Rand, 1990*; *Simmons, Bertolotto & Narins, 1992*). Further, given that at dominant frequencies, the female auditory system is maximally receptive and the calling stimulus triggers the female hormonal response (*Gerhardt, 1974*), observation of multiple bands in a species and their potential functions in species recognition is of particular interest. Within species, dominant frequency has been shown to be affected by social interactions, resulting in males altering their frequency or call behaviour when calls overlap with neighbouring males (*Lopez et al., 1988*; *Wagner, 1989*; *Bee & Perrill, 1996*; *Howard & Young, 1998*; *Bee, Perrill & Owen, 2000*). While the significance of these unusual calls in *M. nilphamariensis* is not known presently, future playback experiments could help test their potential functions in different contexts.

Call variability may arise due to morphological, environmental or social factors and is important to completely understand signal evolution and sexual selection in anurans (*Howard & Young, 1998*; *Tanner & Bee, 2019*). A central aim of studies that describe anuran vocal repertoire is to identify sources of signal variation and their potential for individual discrimination (*Bee & Gerhardt, 2001*; *Bee, 2004*; *Bee & Micheyl, 2008*; *Feng et al., 2009*; *Tanner & Bee, 2019*). To this end, we examined variability within and among individuals. Spectral properties were the least variable of all measured properties: With average among-individual (CVa) values ranging from 3.4 to 7.2 and within-individual values (CVw) between 0.7 and 2.2, dominant frequencies DF 1 and DF2 showed the least variability (Table 3). Among temporal properties, pulse rate had the lowest variability both among and within individuals. As stated earlier, these would be classified as static, consistent with previous studies of anuran vocal behaviour (*Bee, Suyesh & Biju, 2013b*; *Thomas et al., 2014*; *Tanner & Bee, 2019*). In this study, while DF1 was mostly constant ($\overline{X}$ = 1.6 ± 0.1), DF2 was more variable ($\overline{X}$ = 3.5 ± 0.3) across males and the call overall dominant frequency (ODF) shuffled between these peaks (Table 2). These observations could help design future playback experiments to test the role of different frequency bands in this species, *i.e.,* DF1 could function more towards species recognition and subject to stabilizing selection, while DF2 may function towards introducing some variability and function in weakly directional selection under biophysical constraints (*Bee, Suyesh & Biju, 2013b*). The production of these frequencies is dictated by the size of vocal chords within a species' range: Larger males have larger vocal cords and can produce sounds with slightly lower frequencies. When considering ratios of among-individual to within-individual variability (CVa:CVw), the highest ratios were also obtained for these properties, *i.e.,* DF1, DF2 and pulse rate. Hence they hold potential for individual discrimination. Since the values of the frequency remained mostly constant in any given individual, differences in

size between individuals explains the higher CVa values for this property. Similarly, among 'static' temporal properties, pulse rate has been described as a signal of species identity (*Gerhardt, 1991*; *Tanner & Bee, 2019*), which explains its low variability. Higher CVa values are likely dictated by differences in energy reservoirs across males. We obtained intermediate to high values of CVw for most temporal properties, in line with the general observation that dynamic properties tend to vary more with the immediate calling environment and are under strong directional selection towards extreme values (*Gerhardt, 1991*; *Ryan & Keddy-Hector, 1992*; *Gerhardt & Watson, 1995*). Notably, for some of these properties such as call duration, call rise time, pulses/call, pulse fall time, pulse 50% fall time *etc.* we obtained high CVa:CVw values (*i.e.,* >1.3), indicating that these may also function as recognition cues (*Robisson, Aubin & Bremond, 1993*; *Jouventin, Aubin & Lengagne, 1999*).

## Correlation analysis (*M. nilphamariensis*)

Biophysical constraints on calling were evident when interpreting correlation between call properties and parameters such as SVL, temperature and body condition. We found significantly negative correlations for spectral properties: call DF1 with SVL, weight and body condition; while pulse DF1 with SVL and weight. While SVL and weight are direct indicators of body size, body condition indicates the degree of 'fatness' or length-independent-mass for an individual (*Baker, 1992*; *Howard & Young, 1998*). Such size-related information could be conveyed by spectral content of advertisement calls, thus can potentially influence female mate choice (*Ryan & Keddy-Hector, 1992*) as well as male-male competition (*Davies & Halliday, 1978*). The observation that only DF1 was found to be significantly correlated with predictors of body size and condition, indicates that the production of this component is directly dependent on vocal cord size, suggesting that DF1 could be under stabilising selection, while DF2 could have an 'accessory' role in calling behaviour, as suggested above. Further, body condition can also be taken as a proxy for the amount of resources at an individual's disposal to invest in calling (*Schulte-Hostedde, Millar & Hickling, 2001*). Hence, individuals with higher body condition would be expected to call for longer durations (*Podos, 1997*; *Tanner & Bee, 2019*). Our study indicates that in *M. nilphamariensis*, there are significantly positive correlations of body condition with temporal properties such as call duration, call rise time, etc, thus supporting the 'motor performance hypothesis', according to which body condition has positive influence on the ability of males to perform energy-costly activities, like vocalization, in a repeated and consistent manner (*Podos, 1997*; *Tanner & Bee, 2019*).

Finally, although frogs are ectothermic and ambient temperature considerably influences their calling behaviour (*Wells, 2007*), we did not obtain significant correlations with temperature, except for first pulse period, which was correlated negatively with wet bulb temperature. We obtained marginally significant positive correlations with call duration, pulse rate, pulses/call (nearly significant, $p = 0.06$), call duration and call rise time, in agreement with other studies (*Bee, Suyesh & Biju, 2013b*), while all other temporal properties were negatively correlated (not significant) with the wet bulb temperature (Table 3). The lack of stronger correlations that would be otherwise expected with temporal

properties can be accounted for by considering the relatively narrow range of temperature within which recordings were made, *i.e.,* between 26.7 °C and 29.8 °C.

## Comparison of vocal repertoire between *M. nilphamariensis* and *M. ornata* and geographic variation within *M. ornata* populations

Studying call variation is important to reveal intra- and inter-specific patterns of acoustic signals and unravel the processes involved in their generation, maintenance and evolution. Recently, *M. ornata* and *M. nilphamariensis* were reported to belong to a common species group with considerable genetic divergence as well as strong population structures within these species (*Garg et al., 2018*). Our results from analyses of their vocal repertoire indicate inter-specific acoustic distinction as well as within-species differentiation for the two *M. ornata* populations (Fig. 4). Much of this distinction can be attributed to static properties such as call dominant frequencies and pulse rate, as indicated by PCA and DFA results (Table 5). As discussed in the previous section, static properties, particularly spectral parameters are often important for species recognition and reported as reliable signals for within-individual discrimination (*Bee & Gerhardt, 2001*; *Bee & Gerhardt, 2001*; *Bee, 2004*). Generally, individuals are constrained to exhibit call behaviour within a range that prevents maladaptive hybridization, yet allows signal variability (*Servedio & Noor, 2003*; *Rodríguez-Tejeda et al., 2014*). Hence, for multi-component signals such as advertisement calls, it is suggested that some elements function in species identity while others indicate male quality (*Candolin, 2003*). Within this framework, our results suggest that static properties such as dominant frequencies and pulse rate may help capture acoustic differentiation between and within species (*Narayanan, Suyesh & Das, 2021*). Recent bioacoustic comparisons of *Microhyla* species also imply the role of static properties such as dominant frequency and pulse rate to highlight differences between sympatric species as well as within populations of a single species (*Wijayathilaka et al., 2016*; *Wijayathilaka & Meegaskumbura, 2016*; *Chen et al., 2020*). Differences in dominant frequency have been found to be a major factor in phonotaxis experiments on female discrimination between local *vs.* foreign males as well as in explaining genetic divergence and reduced gene flow among distant conspecific populations, further highlighting their potential in the evolution of divergent female preferences and incipient speciation (*Ryan & Wilczynski, 1991*; *Boul et al., 2007*; *Funk, Cannatella & Ryan, 2009*).

Analyses based purely on acoustic descriptions are only the first step to disentangle the complex interplay between factors that give rise to geographic variation in mating signals. These include ecological factors that determine call transmission and may lead to local adaptation in populations (*Wilkins, Seddon & Safran, 2013*; *Velásquez et al., 2013*), morphology, *e.g.,* body size (*Ophir, Schrader & Gillooly, 2010*), genetic drift (*Irwin, Thimgan & Irwin, 2008*; *Lee et al., 2016*) in concert with sexual selection (*Boul et al., 2007*). Studies on geographic variation of signals often investigate correlation between genetic, geographic and acoustic divergence. While contrasting results have been obtained both in support and a lack of association between these aspects (reviewed in *Velásquez, 2014*), an emergent understanding is that behavioural changes (for *e.g.,* acoustic signals) precede genetic changes and call divergence could occur much more rapidly due to local

habitat differences (*Pröhl et al., 2007*). This is particularly relevant to our target *M. ornata* populations from the western coast and Sri Lanka, as the acoustic distinction captured in our analysis could be such an instance. Detailed analyses of potential causes of apparent acoustic divergence are needed, for *e.g.*, targets of selection on local factors such as body size, *etc.* which could account for geographic variation in call properties (*Gerhardt, 1991*; *Pröhl et al., 2007*), along with experiments on female preferences. Hence, despite obtaining statistical differences between populations based on their call behaviour, such results should be treated with caution (*Rivera-Correa, Fernando & Taran, 2017*). Simultaneously, they also invite a number of questions on call variation and signaling ecology in microhylids, such as variation in advertisement calls across other *M. nilphamariensis* populations, studying *M. ornata* populations throughout southern India and examining correlation between genetic and acoustic divergence, especially for allopatric populations from mainland India and Sri Lanka.

# CONCLUSION

In this study, we describe the vocal repertoire of *M. nilphamariensis* from Delhi, consisting of advertisement calls with two other call types and discuss the patterns and sources of call variability. Individuals from this species possibly represent the smallest vertebrate found in this region, highlighting the need for extensive sampling of herpetofauna in urban cities and raising awareness about threats to their existence.

Further, comparative analyses of vocal behaviour with *M. ornata* indicate acoustic distinction between the two species, as well as between *M. ornata* populations from the western coast and Sri Lanka. Clearly, bioacoustics studies are crucial not only for integrative taxonomy (*Garg et al., 2021*), but could be valuable in generating testable hypotheses on signal variation, evolution and divergence among cryptic species (*Bedi et al., 2021*). At the same time, the impacts of such studies for conservation strategies are of immediate environmental concern.

**Abbreviations**

Acronyms and abbreviations for frequently used terms are as follows:

| | |
|---|---|
| **WC** | Western Coast |
| **SL** | Sri Lanka |
| **MS** | Megha Srigyan |
| **AS** | Abdus Samad |
| **ABS** | Abhishek Singh |
| **JK** | Jyotsna Karan |
| **AC** | Abhishek Chandra |
| **PGS** | Pooja Gokhale Sinha |
| **VK** | Vineeth Kumar |
| **SD** | Sandeep Das |
| **AT** | Ashish Thomas |
| **RS** | Robin Suyesh |

## ACKNOWLEDGEMENTS

RS, MS, AS, ABS, JK, AC, PGS are grateful to (late) G.P.C. Rao, Department of Botany, Sri Venkateswara College, University of Delhi for his guidance and support on designing this project. We thank Mark A. Bee for guidance and helpful discussions on acoustic communication and analyses, Madhava Meegaskumbura and Nayana Wijayathilaka for releasing raw acoustic recordings of *M. ornata* (published in *Wijayathilaka & Meegaskumbura (2016)*) and P. Hemalatha Reddy, Former Principal, Sri Venkateswara College, University of Delhi and K S Rao for providing support and guidance. RS would like to thank Kanishka DB Ukuwela for providing photographs of *Microhyla ornata* from Sri Lanka and to Amit Vashishtha, K.V.Gururaja and Duminda Dissanayake for providing logistical support for the study. RS, SD and AT would also like to thank Lilly Eluvathingal Linden for her support in providing the field equipments. We would also like to thank the Delhi Development Authority (DDA) personnel for their support while performing fieldwork.

### Funding

This study was carried out as part of the Innovation Project SVC-304 to Robin Suyesh, Abhishek Chandra and Pooja Gokhale Sinha under the Scheme of Innovation Projects 2015-16, funded by the University of Delhi. Megha Srigyan, Abdus Samad, Abhishek Singh and Jyotsna Karan were supported as Innovation Project fellows. There was no additional external funding received for this study. The funders had no role in study design, data collection and analysis, decision to publish, or preparation of the manuscript.

### Grant Disclosures

The following grant information was disclosed by the authors:
The University of Delhi.

### Competing Interests

The authors declare there are no competing interests.

### Author Contributions

- Megha Srigyan performed the experiments, participated in fieldwork, analyzed the data, prepared figures and/or tables, prepared the manuscript, authored and reviewed drafts of the paper, and approved the final draft.
- Abdus Samad performed the experiments, participated in fieldwork, analyzed the data, prepared tables, reviewed drafts of the paper, and approved the final draft.
- Abhishek Singh performed the experiments, participated in fieldwork, analyzed the data, reviewed drafts of the paper, and approved the final draft.
- Jyotsna Karan performed the experiments, participated in fieldwork, analyzed the data, reviewed drafts of the paper, and approved the final draft.
- Abhishek Chandra participated in fieldwork, contributed reagents/materials/analysis tools, reviewed drafts of the paper, and approved the final draft.

- Pooja Gokhale Sinha participated in fieldwork, contributed reagents/materials/analysis tools, reviewed drafts of the paper, and approved the final draft.
- Vineeth Kumar performed the experiments, participated in fieldwork, contributed reagents/materials/analysis tools, reviewed drafts of the paper, and approved the final draft.
- Sandeep Das performed the experiments, participated in fieldwork, contributed reagents/materials/analysis tools, reviewed drafts of the paper, and approved the final draft.
- Ashish Thomas performed the experiments, participated in fieldwork, contributed reagents/materials/analysis tools, reviewed drafts of the paper, and approved the final draft.
- Robin Suyesh conceived and designed the experiments, performed the experiments, participated in fieldwork, analyzed the data, contributed reagents/materials/analysis tools, prepared figures and/or tables, prepared the manuscript, authored and reviewed drafts of the paper, and approved the final draft.

## Animal Ethics

The following information was supplied relating to ethical approvals (i.e., approving body and any reference numbers):

The study was conducted outside protected area (public place) and none of the animals were collected during the study so study permits were not required.

## Data Availability

The raw data are available in the Supplemental Files.

## Supplemental Information

Supplemental information for this article can be found online at http://dx.doi.org/10.7717/peerj.16903#supplemental-information.

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
