# Peer review of "Vocal repertoire of Microhyla nilphamariensis from Delhi and comparison with closely related M. ornata populations from the western coast of India and Sri Lanka"

_PeerJ, doi:10.7717/peerj.16903_

## Round 0.1 · original submission · Minor Revisions

Please revise your manuscript to address the concerns of the reviewers.

Reviewer 1 ·

Basic reporting

The paper is clearly written and well-argued. I find no flaws in the arguments. Authors have presented sufficient data and files as supplements.

Experimental design

Experiments have been conducted rigorously and are correctly described throughout the manuscript. Methods described with sufficient details and results supported the entire argument. The paper is an original and important contribution to herpetological studies.

Validity of the findings

The research described in the manuscript is technically sound and data strongly support the conclusions provided by the authors. The sample size and analysed material are adequate, well presented. The manuscript is statistically sound and correctly done.

Additional comments

I recommend accepting this paper for publishing after minor corrections.

A few minor points need comment and/or minor changes:

1. In the abstract, I suggest including more details of analysis results. As an example, please mention some important values of call properties and variations.

2. I would like to suggest authors focus on minor mistakes such as Figure 3 as an example. In Figure 3 caption’s details contain marking “B), C), D)”, but “A)” is missing at the beginning.

Reviewer 2 ·

Basic reporting

No negative comment.
The authors present a concise and very clearly-written manuscript. Their purely exploratory objectives are well framed with relevant anuran and acoustic literature, and the utility and importance of such research is well laid out. The introductory figures and tables unambiguously prepare a non-expert reader for their analysis. The results are illustrated with professional, and (mostly) clearly captioned figures and tables, with sufficient reference in the methods, results, and discussion. Supplemental and raw data are provided in clear, user-friendly formats, and high quality video and audio examples of the study species are available as well. As a whole, the study stays consistent with its expressed purpose, and discusses the results in a straightforward manner that logically follows from the introduction and objective, with key suggestions for further research in a now more robustly characterized clade of chorusing anurans.

Experimental design

No negative comment.
This project applied well-established methodologies to an under-studied group of closely related species, and seems completely in line with the purpose of PeerJ. In addition to quantifying the male advertisement vocalizations of a novel species using a multitude of distinct acoustic parameters, the authors then investigate sources of variation and test the discriminability of this call repertoire in comparison with two populations of a close heterospecific. The data collection and analysis are rigorously described, and could straightforwardly be used as a methodological template for future investigative work in other taxa. Comments and thoughts on some details of the methods and analysis are laid out in General Comments below, but none are fundamental enough to call into question the experiment as designed and performed.

Validity of the findings

No negative comment.
The authors make note of the results which are generally consistent with similar research in more widely studied anurans (e.g. the “static” and “dynamic” features of temporal/spectral call traits), and do so with detailed reference to that literature. They also highlight the stark acoustic divergence that was quantified among the heterospecific populations; that DFA of acoustic data can assign two populations of M ornata to geographic population with exact accuracy (given samples as small as 5) is indeed noteworthy. Their conclusions do not overstate any of the findings, which provide crucial background information for subsequent mate-choice research, and they prompt further investigation into both shape of selection on the mating signal, as well as the function of the various call type classifications in animal communication.

Additional comments

Line 145: As social context is widely known to affect advertisement calls, many readers will expect chorus size to be a potential factor here. I expect that your choruses between Aug-Oct were small, but perhaps briefly mention how variable the chorus attendance was across collection nights. This would probably only be relevant to the population in which you sampled 18 calling males across the season.

Line 200-201: While I agree that the gap between calls is not necessarily itself a trait, it is an important component (combined with call duration) of calling period (or the inverse, calling rate). Intercall interval is the only metric for calling rate present in Table 1, and is shown to correlate generally with body size and condition in Table 3, but is then omitted in later multivariate analyses (Line 239). I would not argue for re-analysis; it would simply be useful to clarify in the methods why larger scale temporal parameters (general calling effort) were omitted. I imagine that the authors are aware of this and omit it because it would be more relevant in a mate choice context than in a species discrimination context (and I agree).

Line 237-238: As this seems to include data for all call types (some of which may not be advertisements), it would be interesting to see how much the PCA would change if it incorporated only the standard Type 1 from M. nilphamariensis. Probably overkill at this early stage, as we do not yet know how each type functions in communication, but might the diversity of calls in the novel species drive some of the observed divergence from the M. ornata samples? (which presumably contain only standard mating calls).

Line 239: Perhaps remind the reader why intercall interval will not be present in Tables 2,4,or 5, but was relevant to the correlations and CoV in Table 3.

Line 264-266: How did you distinguish between an inter-call interval and an inter-bout interval?

Line 300-301: This wouldn’t be necessary, but it seems Repeatability (Dingemanse et al. 2010) could be calculated using the components of variance you employ here – and it would provide a significance statement for “among individual” variation outweighing “withing individual”. Knowing which traits vary repeatably in a population could direct focused sexual selection playback experiments in this taxon.

Line 336: “For example” instead of “For e.g.”

Line 339: typo in frequency

Lin 426-427: Was this only the case for F1 x F3 (as in Fig 4)? Readers may wonder why plots representing the other factors are omitted.

Line 647: Singular ‘concern’ would read better than the plural.

Fig. 1: While a deep dive on coloration would certainly be a distraction in this paper, many readers will note the visual difference between the WC and SL populations of ornata. If these are representative images (provided someone has done some coloration work on this), perhaps a brief mention of this divergence in the intro would be interesting (and would bolster your later findings that their acoustics cluster separately).

Table 2: Caption cuts off at ‘individua’

Table 2: What does the red box with an asterisk signify? It is not denoted in the caption.

Table 4: Same comment as above.

Table 7: typo in the second line (score_in)

·

Basic reporting

The article is clearly written and the English is intelligible with the exception of a small list of typos and cases of odd phrasing:
Line 113: “ This necessitates the need for….” The phrasing is odd.
Line 127: period seems to be missing after “Sri Lanka”
Line 131: the phrasing is odd.
Line 331, Type IV calls differ from each other?
Line 407, “For e.g”, change for “For example”?
Paragraph 406-414 is an important summary of the comparison between species, and I had a hard time reading it. The authors may consider re-writing it in a clearer form.

The background and relevance of the study is justified with the appropriate context and supported with the pertinent references to the literature.

The article is professionally and clearly structured, with a comprehensive description of the methods, a thorough presentation of the analysis performed and the results. The conclusions are clearly linked to the data and results.
Examples of the raw data are included in the supplementary material, but the complete dataset is neither included, nor pointed to as a data repository. Given that this is an ecologically valuable dataset (lines 472-476), I encourage the authors to deposit the full dataset in a repository (such as zenodo or figshare) upon acceptance of the paper, and include a pointer to it in the final, published version of this study.
I commend the authors on the transparency of the presentation: the intermediate steps of the analyses are shared as supplementary materials.
The work is supported by thorough presentation of the data in the form of tables and in a few cases, with figures. Some of the points would be better conveyed by presenting figures accompanying the tables.
Figure 2 sufficiently conveys examples of the temporal structure and spectral contents of the different calls that are later analysed. Yet it could be made clearer with two relatively simple modifications. First, I suggest marking the temporal areas of interest when ‘zooming’ in in time (e.g., when going from 10 seconds of waveform in panel D to 2 sec. in panel G to 0.5 sec in panel J, which segment of the preceding panel is being shown?). Second, the spectrograms could do a better job of illustrating the contents of the corresponding waveform. The width of the spectrogram window seems too large, in particular in panels M, N, leading to a blurry representation. Also, the dynamic range of the spectrogram seems too low as well: note that in panel P, for instance, the energy around the ~20kHz bin is noticeable, but in the corresponding panel M the decay looks sharp after around ~8kHz.

The study is self-contained. In this manuscript, Srigyan et. al collect a comprehensive dataset representative of the vocal behavior of Microhyla nilphamariensis from Delhi, India, paired with observations of the physiology and environment of the individuals. This dataset allows them to perform a quantitative description of the vocal behavior. Furthermore, they can provide a comparison with other species, and discuss the relevance of certain features of the vocalizations in differentiating the calls within and between species. One of the primary aims of the study is to “generate a comprehensive account of the vocal repertoire of M. nilphamariensis in Delhi.”: this both seems relevant and accomplished.

Experimental design

The manuscript fits well within the scope of the journal, at the interface between biological and environmental sciences, as a research article.

The research question is well defined (the description of the repertoire of the M. nilphamariensis in Delhi.), which was not previously described.

The methods are described in detail and, except from the complete dataset, transparency abides high standards.

Validity of the findings

The work is supported by thorough presentation of the data in the form of tables and in a few cases, with figures (but see comments in 1.)

Reviewer 4 ·

Basic reporting

There are a few areas of the Results section that need some gentle copyediting for consistency with regard to abbreviating words versus spelling them out (DF1 and DF2 versus "dominant frequency 1 & 2"), using "and" rather than "&", and including spaces between "M." and specific epithets. In general, I found the article very clearly written and have no other comments about the language.

Literature/references:
-The authors reference Tanner and Bee 2019 (line 569) to support the idea that we would expect body condition to be related to temporal call characteristics like call duration. However, while that paper does address the hypothesis that call traits are related to body condition, it actually goes on to report there is no relationship between body condition and any of the call traits those authors examined. I would encourage the authors to provide enough background/context about this paper to allow the reader to gauge whether it supports the hypothesis proposed.
- Lines 542-543 - It's true that pulse rate is a species recognition signal in gray treefrogs, and also true that this phenomenon of pulse rate being useful for species recognition may occur more broadly in acoustically communicating species that have pulsed signals. I think this sentence may need a bit more in the way of references or explanation to support the idea that pulse rate specifically (or else static traits in general) may broadly function as (a) species recognition signal(s).

Experimental design

The acoustic analyses are in general really detailed and appropriate. The main question I had reading the manuscript is that, in Figure 3, the calls are presented as belonging to different types (1-4) but the way the PCA (Figure 4), the sample sizes for the number of individual calls (lines 347), and the statistics in Table 2 are presented all imply that the analysis grouped these different "call types" together for the multivariate analyses. Because there seem to have been so few calls produced of types 2, 3, and 4, maybe they wouldn't affect the PCA/DFA strongly, but perhaps the authors can address whether they were included or excluded.

Validity of the findings

I think the sample size (n = 18, n = 5, n = 5) are somewhat low to draw robust conclusions in some ways because of the remarkable variation we often see within individuals in anuran vocalizations, even over short timescales. I think the authors have done a good job discussing the amount of within-individual versus between-individual variation they measured and why it matters, but there are some places (line 484) where they perhaps overstate how robust their sample is. In particular, the authors commented on the relatively high variation in M. nilphamarensis in their PCA plot (Figure 4) compared to the two, tighter clusters of M. ornata. But looking at the M. nilphamarensis cluster, I wonder if a subsample of n = 5 males from that group would be fairly likely to randomly generate a tight cluster such as we see with M. ornata. The authors might be able to address that by generating random sub-samples of 5 males from the M. niphamarensis dataset and re-running their analyses (but such an analysis is not necessary to publish the article, in my opinion).

In lines 529-530, the authors discuss the magnitude of variation in DF2 being higher than what is present in DF1, but I was curious about this. If DF2 is harmonically related to DF1, we would expect them to be linearly related within frogs. Spectral traits like bandwidth or frequency-modulation would then have higher magnitudes in DF2 than in DF1. Is the harmonic relationship between these two traits driving this difference in variability? Maybe the authors can explain why this is (or isn't) the case.

Line 334: It seems that call type IV was represented by a single call produced by a single individual, and that call types 2 and 3 were similarly rare.

Additional comments

It seems to me that the behavioral relevance (or irrelevance) of this variation in signaling behavior to the receivers (females) is an important aspect of quantifying differences in signaling behavior, and it would be nice to see that included in the authors' discussion of reasons why statistical differences between these populations should be treated with caution (liens 628-629). Relatedly, are the behavioral functions of the different call types described known?

I have a few comments about figures and tables:
Figure 2: Can the authors add to the caption which call types are presented here?
Table 2: The caption seems to cut off mid-sentence? It's also potentially confusing to include the phrase about the sample of 5 individuals, which conflicts with the (n = 18) earlier in the caption.
Table 4: It's a little odd that the columns are presented in this order (2, 3, 4, 1), though I can see how the authors arrived at this decision.

---

## Round 0.2 · accepted · Accept

Thank you for your careful revisions. I now find the paper acceptable for publication.